# The Time–Temperature Superposition of Polymeric Rubber Gels Treated by Means of the Mode-Coupling Theory

**DOI:** 10.3390/gels10050313

**Published:** 2024-05-03

**Authors:** Domenico Mallamace, Giuseppe Mensitieri, Martina Salzano de Luna, Francesco Mallamace

**Affiliations:** 1Section of Industrial Chemistry, Department of ChiBioFarAm, University of Messina, CASPE-INSTM, V.le F. Stagno d’Alcontres 31, 98166 Messina, Italy; mallamaced@unime.it; 2Department of Chemical, Materials and Production Engineering, University of Naples Federico II, Piazzale Tecchio 80, 80125 Naples, Italy; giuseppe.mensitieri@unina.it (G.M.); martina.salzanodeluna@unina.it (M.S.d.L.); 3Department Matematica, Fisica, Informatica e Scienza della Terra, University of Messina and Istituto Sistemi Complessi del CNR, 00185 Rome, Italy

**Keywords:** viscoelasticity, relaxations, polymeric gels, dynamical arrest

## Abstract

Viscoelastic relaxation measurements on styrene-butadiene rubbers (SBRs) doped with carbon nanotube (CNT) at different concentrations around the sol-gel transition show the time–temperature superposition (TTS). This process is described in terms of the mode coupling theory (MCT) approach to viscoelasticity by considering the frequency behavior of the loss modulus $E^{\prime \prime }\left(\omega \right)$ and showing that the corresponding TTS is linked to $\omega^{1/2}$ decay. From the analysis of the obtained data, we observe that the interaction between SBRs and CNT determines different levels of decay according to their concentration. Systems with the lowest CNT concentration are only characterized in the studied *T*-range by their fragile glass-forming behavior. However, at a specific temperature $T_{L}$, those with the highest CNT concentration show a crossover towards pure Arrhenius that, according to the MCT, indicates the presence of kinetic glass transition (KGT), where system response functions are characterized by scaling behaviors.

## 1. Introduction

Liquids approaching transitions characterized by dynamic slowing down, like sol-gel (typical of polymeric systems and solutions) or liquid glass, have very long relaxation times and very high viscosities compared to room-temperature water. The investigation of these highly viscous materials is fascinating because “independent of their chemical nature of the intermolecular bonds” they share a number of common features [1,2,3,4,5,6,7,8,9,10,11,12,13]. The most significant are the non-Arrhenius temperature dependence of the average relaxation time (before the transition), non-Debye linear response functions, energy landscape configuration, and a fragile (super-Arrhenius) to strong (Arrhenius) dynamic crossover (FSDC), i.e., a change from Vogel to Arrhenius behavior at the temperature $T_{L}$, observed by decreasing *T* and located before the dynamic arrest or transition glass temperature $T_{g}$.

As it is well known, the FSDC characteristic of glass-forming materials accompanied by the breakdown of the well-known Stokes–Einstein relationship marks an absolute change in the system dynamic properties [10,13]. Whereas the Vogel region is dominated by translational motions, below $T_{L}$ are the relevant effects of dynamic heterogeneities and hopping processes [9,12]. For the study of these peculiar chemical–physical conditions, theoretical models based on scale concepts have proven effective [5,6,7]. The percolation [5] and mode-coupling theory (MCT) [9] are certainly the most used.

Linear response experiments are usually reported in terms of frequency dependence (or alternatively the time) of the imaginary part (loss contribution) of the response function (susceptibilities) χ(ω). When the shape of the loss peak in a log–log plot is *T* independent, the liquid obeys the time–temperature scaling principle (TTS or TTS principle) with respect to the response function. Mathematically, if τ(T) is the average relaxation time, TTS is obeyed whenever functions *N* and ϕ exist such that χ(ω,T)=N(T)ϕ[ωτ(T)].

It has been established that the TTS principle represents a useful tool for estimating changes in properties of polymer materials at long intervals or extreme temperatures. In particular, the viscoelastic behaviors of these systems show both time and temperature dependences not only above the glass temperature $T_{g}$ but also below. Basically, the TTS principle is related to the concept of equivalence between *T* and τ(T); under these conditions it is possible to establish simple temperature functions that enable us to translate isothermal segments of the considered response function (e.g., the elastic modulus) along the time scale and compose a master curve, registered as a reference temperature ($T_{ref}$) [14].

When the TTS principle applies, response functions are easily determined over many decades of frequency and the procedure works even if only some ω decades are experimentally accessible. The *T*-shifting function, also called the shifting factor, can be described by the Williams–Lendel–Ferry (WLF) equation for systems near and above $T_{g}$ [15,16] or the Arrhenius-type equation for those below it ($T<T_{g}$) [14]. It should be noted that the WLF is equivalent to the Vogel–Fulcher–Tamman–Hesse equation (VFTH) [17,18,19].

Despite the fact that the TTS principle has been used intensively over the years, it is not universal. TTS violations are not just exceptions for molecular liquids but are also quite common. It was initially assumed that these violations were due to the interference effects between relaxation processes (e.g., β and α), with the β relaxation often found at frequencies higher than those of the dominant α relaxation. Nowadays, it is established that the α process usually violates the TTS principle. Although it has been used continuously in the study of polymers, for these reasons, there has been little interest in the TTS principle for non-polymeric viscous liquids close to $T_{g}$.

However, there is a strong connection between the TTS and the ideal MCT, which is widely used by the scientific community to explain the dynamics and relaxation functions of glass-forming materials in terms of accurate scaling laws. In less viscous regimes at higher temperatures, where MCT is believed to apply, this theory’s TTS prediction has received some confirmation from both experiments and computer simulations [9]. From the MCT perspective, breakdowns of TTS in the very high viscosity regime correlate with the well-known breakdown of ideal MCT formalism.

This paper addresses the behavior of highly viscous liquids (near $T_{g}$) and the motivation for reinvestigating the validity of TTS in this regime by integrating it with MCT approaches for viscoelasticity. In particular, we have followed a suggestion proposed by a study conducted on molecular liquids, dedicated to verifying the general validity of the TTS in thermodynamic conditions, e.g., determining whether the two main relaxations (α and β) have an influence on each other [20]. Furthermore, this study, dealing with the data of supercooled triphenyl phosphate (and other molecular liquids close to the calorimetric glass transition), shows that TTS is fully obeyed for the primary relaxation process and is linked to an $\omega^{-1/2}$ high-frequency decay of loss (and wider frequency range on the right side of the relaxation peak). Therefore, it seems that, when the TTS holds, this last result is general, as predicted by some related theories [21,22].

Starting from these observations, we were motivated to provide a new contribution to the TTS investigation in the present work using extended MCT formalism [23]. This analysis of dynamic mechanical analysis data (DMA) characterizes the viscoelastic behavior of cross-linked styrene-butadiene rubbers filled with carbon nanotubes in the gel phase versus $T_{g}$ and for the region *T*<$T_{g}$. The basic reason for the following study is that the extended version of the MCT [23,24] rather than the ideal one, and the TTS, is capable of describing physical processes for temperatures lower than those of dynamic arrest. As is known, under these conditions, the system properties are characterized by dynamic heterogeneities and hopping processes. Therefore, only by taking these specific peculiarities into account can an appropriate description of this material be possible. The data we analyze here have been previously published in cooperation with the National Institute of Industries and Technology (INTI, Argentina) [25]. The reader is referred to [25] for full experimental details. Furthermore, it should be noted that CNTs have a remarkable influence on the physical/chemistry properties of polymeric materials, including viscoelasticity and thermal characteristics. A tensile strength test was selected to evaluate the complex modulus E.

The first step was to consider the MCT basic properties in the description of viscoelastic processes. After considering how this model describes the relaxation processes highlighted by the system response functions (susceptibility or specifically the shear modulus $E^{\prime \prime }\left(\omega \right)$), we then considered the idea of choosing basic parameters for the TTS as the minimum response function ($E_{min}^{\prime \prime }$) and its frequency value $\omega_{min}$, both of which were obtained by proper MCT data fitting for each concentration and temperature. In this way appropriate TTS master curves were obtained for each CNT concentration of the studied SBRs. The analysis of the properties of these suggests that the interactions between carbon nanotubes and styrene-butadiene rubber strongly influence the chemical–physical properties of the entire system, determining its basic thermodynamics.

The temperature behaviors of $E_{min}^{\prime \prime }$ and $\omega_{min}$ show that, while the rubbers with the lowest CNT concentration are only characterized in the studied *T*-range by their fragile glass-forming behavior, those with the highest concentration show an FSDC crossover at a specific $T_{L}$.

In terms of the MCT, this provides evidence that the system, under these conditions, is characterized by kinetic glass transition due to the effects of the CNTs on the polymeric rubber. In other words, the presence of carbon nanotubes shifts the $T_{g}$ of the mixture towards higher temperatures than that of the pure, basic system.

## 2. Viscoelasticity Models

As demonstrated in polymer science, the system viscoelasticity is well characterized by the dynamic complex viscosity $\eta^{*}\left(\omega \right)$ and the complex shear modulus $G^{*}\left(\omega \right)$ (or the compliance $J^{*}\left(\omega \right)=1/G^{*}\left(\omega \right)$) [15]: $G^{*}\left(\omega \right)=G^{\prime }+iG\prime \prime =i\omega \eta^{*}=i\omega \left(\eta^{\prime }+i\eta \prime \prime \right)$. $G^{\prime }$ and G″ are the loss and storage (or elastic) moduli, respectively, and $\eta_{0}=Lim_{\omega \to 0}G\prime \prime /\omega$ is the zero frequency viscosity. In oscillatory experiments, such as the one performed here, these moduli are obtained from the measurement of the time dependence of the stress σ as follows:(1)$$\sigma =\gamma (G^{\prime }\sin\left(\omega t\right)+G^{\prime \prime }\cos\left(\omega t\right)),$$
where γ is the strain amplitude. The equation showing that the stress amplitude varies as $\sigma =\sigma_{0}\sin\left(\omega t+\delta \right)$ also proposes the connection between the stress/strain phase angle δ(ω) and the moduli as $G^{\prime }=\left(\sigma_{0}/\gamma \right)\cos\delta$ and $G^{\prime \prime }=\left(\sigma_{0}/\gamma \right)\sin\delta$ so that it is $G^{\prime \prime }/G^{\prime }=\tan\delta$.

In addition, shear (*G*) and the other elastic moduli (*E*, Young’s and *K*, bulk moduli) are related by means of the Poisson’s ratio: υ=E/2G−1 [15]. Considering the previous discussions in terms of system susceptibility, thermodynamic behavior can be described in terms of the MCT (in both the ideal and extended forms).

### 2.1. Mode Coupling Theory

In both the ideal [9] and extended [23,24] forms developed using cage effects, it is assumed that the time-dependent shear viscosity can be written in terms of the structure factor S(q) (the space Fourier Transform of the local density correlation function g(r)) as follows [26]:(2)$$\eta \left(t\right)=\frac{k_{B}T}{60\pi^{2}}\int_{0}^{\infty }dq{\left[nq^{2}\frac{dlnS(q)}{dq}\varphi \left(q,t\right)\right]}^{2},$$
with the complex shear modulus $G^{*}\left(\omega \right)$ given by the corresponding Laplace transform:(3)$$G^{*}\left(\omega \right)=i\omega \int_{0}^{\infty }dte^{i\omega t}\eta \left(t\right).$$

When close to the kinetic glass transition (KGT) and the lowest order in the control parameter σ (such as ρ, *P*, or *T*; i.e., $\sigma =|T-T_{c}/T_{c}|$), the $G^{\prime \prime }$ behaves like the susceptibility by showing extremes (minima and maxima) and scaling properties. According to this, the MCT equation holds for all correlators between variables which have an overlap with density fluctuations and depend singularly on two scales of σ and time, respectively, of an amplitude or correlation scale $c_{\sigma }={\left|\sigma \right|}^{1/2}$ and a time scale $t_{\sigma }=t_{0}{\left|\sigma \right|}^{\gamma }$.

The exponent γ is γ=(1/2a)+(1/2b), with 0<a<0.5 and 0<b<1 (both non-universal) determined by the so-called exponent parameter $\lambda =\Gamma^{2}\left(1-a\right)/\Gamma \left(1-2a\right)=\Gamma^{2}\left(1+b\right)/\Gamma \left(1+2b\right)$.

$G^{\prime }$ data are usually not accurate enough to allow fitting with MCT; therefore, the measured values of $G^{\prime \prime }$ are considered using the amplitude parameters $G_{1}=g_{1}c_{\sigma }$ and $G_{2}=2g_{2}c_{\sigma }/(\lambda g_{1}$), the frequency $\omega_{\sigma }$, and the exponent parameter λ. For the MCT, scaling in the correlation region implies the scaling of frequency and the loss modulus at the minimum. In particular, the $\omega_{min}$ position and the intensity of $G_{min}^{\prime \prime }$ depend sensitively on σ; thus, it is easy to express the two scales, $1/t_{\sigma }$ and $G_{1}$, in terms of $\omega_{min}$ and $G_{min}^{\prime \prime }$ so that we also expect $\omega_{min}\approx {\left|\sigma \right|}^{1/2a}$ and $G_{min}^{\prime \prime }\approx {\left|\sigma \right|}^{1/2}$.

These susceptibility minima appear in the left side of Figure 1, where the real and imaginary parts tensile moduli are reported, measured by means of the DMA in the polymer of our interest: SBR rubber filled with carbon nanotubes at a concentration of C=5.

Dielectric relaxation and depolarized light scattering experiments have suggested a useful form for the data analysis of $\chi^{\prime \prime }\left(\omega \right)$. A form with which a MCT master curve can be found as follows:(4)$$G\prime \prime \left(\omega \right)=G{\prime \prime }_{\min}\left[b{\left(\omega /\omega_{min}\right)}^{a}+a{\left(\omega /\omega_{min}\right)}^{-b}\right]/\left(a+b\right).$$

This interpolation form (IF) is equivalent to the main MCT form for time correlators [26]. This is precisely the expression used to study the measured tensile loss moduli ($E^{\prime \prime }$) to obtain an estimate of both $\omega_{min}$ and $E_{min}^{\prime \prime }$ as the values of exponents *a* and *b*. An example of the corresponding data fitting is proposed on the top of the right side of Figure 1.

It must be stressed that scaling laws used to describe viscoelasticity at the sol-gel transition threshold (or percolation) are based on the exponents *s*, *t* and Δ=t/(s+t) [27,28,29]. The static shear viscosity and the corresponding elastic modulus (the monomers modulus at some microscopic time scale $\tau =1/\omega_{0}$) are thus defined as follows: $\eta_{0}∼\varepsilon^{-s}$ and $G_{0}∼\varepsilon^{-s}$. Near the threshold, $G^{\prime }\left(\omega \right)$ and $G^{\prime \prime }\left(\omega \right)$ have the following dependence: $G^{\prime }\left(\omega \right)∼G\prime \prime \left(\omega \right)∼\omega^{\Delta }$. In addition, near and above the threshold, the frequency power law of $G^{*}$ has a remarkable consequence: $G\prime \prime /G^{\prime }$ has a universal critical value. More precisely, in rheology, the loss angle δ is defined as $\tan\delta =G^{\prime \prime }/G^{\prime }$ so that one has $\delta_{c}=\left(\pi /2\right)\Delta$. For the MCT, Δ=0.7 is an universal exponent.

### 2.2. The TTS–WLF Approach

Shift factors above $T_{g}$ obtained during the construction of TTS master curves were successfully described by the WLF equation [16]:$$\log S_{F}=\log\frac{\tau (T)}{\tau (T_{ref})}=-\frac{C_{1}\left(T-T_{ref}\right)}{C_{2}+\left(T-T_{ref}\right)},$$
where factors $C_{1}$ and $C_{2}$ depend on $T_{ref}$ and the material. In an expression that usually holds for polymers over the range $T_{g}<T<T_{g}+100\;K$, when $T_{ref}$ is identified with $T_{g}$$C_{1}$ and $C_{2}$, “universal” values close to 290.44 and 324.6 K, respectively, are assumed [16]. As mentioned, this equation is equivalent to the VFTH [17,18,19]:$$\tau \left(T\right)=\tau_{0}exp\left(\frac{B}{T-T_{0}}\right)\;\;\;\;\;\;\;\;\;\;\;\;\;\;\;\;\;\;T_{0}<T_{g},$$
where $\tau_{0}$ is a pre exponential factor and *B* and $T_{0}$ are adjustable parameters. These two models (WLF and VFTH) are the most familiar for describing non-Arrhenius behavior, and their parameters are related by $C_{1}=B/2.303\left(T_{ref}-T_{0}\right)$ and $C_{2}=T_{ref}-T_{0}$.

The region with $T>T_{g}$ where relaxations are characterized by pure Arrhenius behavior, is named the strong glass one. It has time correlations described by a single exponential because relaxation take places between two fixed energetic levels. Instead, for $T<T_{g}$ named the fragile glass forming region, relaxations are non-Arrhenius and, from an energetic point of view, the corresponding physics is much more complex. In that case, the measured correlation functions are characterized by multi-relaxations occurring in a large distribution of energy configurations. In this condition it is also customary to describe the relaxation processes by means of stretched exponential forms.

## 3. Results and Discussion

The material studied here consists of a “doped” styrene-butadiene rubber (SBR rubber) [25]. In particular, it is considered a rubber filled with carbon nanotubes (CNT) at different concentrations with 40 phr carbon black (CB). The relative CNT concentration (*C*) is reported as phr-parts per hundred rubber (phrCNT). Their viscoelasticity was measured through a series of DMA experiments evaluating the complex modulus E via a series of tensile strength tests in the following parts: the shear $E^{\prime \prime }$, the loss $E^{\prime }$, and the relative loss angle δ (see [25] for details).

These polymers behave as highly viscous liquids and are characterized by both calorimetric glass and sol-gel transitions. Therefore, experiments were carried out within a temperature range of 233<T<303 K (with successive steps of 5 K and a thermal stability of 0.1 K), with a frequency interval of 0.8<ω<100 ($\sec^{-1}$). A strain elongation of 1% was used for all the measurements, analyzing the following nanotube concentrations: SBR-0.5phrCNT, SBR-1phrCNT, SBR-5phrCNT, SBR-10phrCNT, and SBR-40phrCB. Using the WLF approach for the data measured at these concentrations (similar in evolution to that illustrated for C=5 in the left side of Figure 1), a correct TTS was obtained. In particular, for the pure rubber, the obtained values were as follows: $T_{0}=273$ K for factor $C_{1}=6.6785$, with $C_{2}=347$ K.

Considering that the susceptibility spectra χ(ω) of the relaxing system are characterized by peaks with maxima and minima, we adopted the MCT method for the TTS using parameters *a* and *b* to obtain $\omega_{min}$ and $E_{min}^{\prime \prime }$ from the fitting of the isothermal spectra by means of Equation (Equation 4). We wanted to obtain more details on the system’s chemicals–physics from its viscoelastic behavior compared to those obtained with the classical methodology based only on WLF formalism. Hence, the TTS was obtained in multiplicative terms by considering the measured temperature shifts of $\omega_{min}$ and $E_{min}^{\prime \prime }$. The results of this procedure are shown in Figure 1 (open symbols) as a log–log plot for the concentration C=5. The figure also shows the TTS of the storage modulus (full symbols) obtained using the same values of $E^{\prime \prime }$ data fitting. This represents additional information considering that the TTS only applies to the shear. As can be seen, the frequency range of the master curves covers seven orders of magnitude in ω and about two in $E^{\prime \prime }$.

Figure 2 shows, as a function of ω, all the master curves obtained through this procedure for all the different studied concentrations (C=0.5,1,5,10 and 40). Visual analysis of these curves shows a possible power law in the high frequency region before the maximum for the loss function. Furthermore, there are concentration-dependent vertical shift factors.

Molecular liquids dielectric analyses for the primary relaxation process near $T_{g}$ show that the TTS of $E^{\prime \prime }$ is linked to $\omega^{-1/2}$ decay [20]. In addition, the presence of an extra background contribution (related to *C* and named $E_{B}^{\prime \prime }$) was also observed in the obtained master curves. In our case, its strong dependence on the CNT concentration suggests an interaction process associated with crosslinking between the polymers of the system due to the increasing development of mesh sizes in the related fluctuations. Fluctuations influence the valuesof the elastic modules at the lowest frequencies. Assuming that the corresponding value is the asymptotic one of $E^{\prime \prime }$ at low ω, we subtracted it from the original data, obtaining $E_{back}^{\prime \prime }=E^{\prime \prime }-E_{B}^{\prime \prime }$ (see Figure 3) as the complete power law behavior for approximately six orders of magnitude in ω.

The final result is that the corresponding index within the experimental errors is just 0.5. Such a result seems to suggest the general validity of the extended MCT analysis of the TTS process. The inset of Figure 3 reports (in a lin–log plot) the dependence of the $E_{B}^{\prime \prime }$ on concentration (*C*). The continuous curve is fit with a logistic function indicating an underlying onset and growth mechanism dependent on the amount of filler. The corresponding flex point is at approximately C∼6. We recall that, in the present case, $E_{B}$ represents the zero frequency shear viscosity or a quantity directly related to it. The data of this inset show how $\eta_{0}$ develops with *C*: for the lowest concentrations of carbon nanotubes, C<1, it is practically constant, then it grows and changes rate at C∼6, and finally, it saturates for C≥15. On the basis of this result, by considering these $E_{B}$ values, we can group the different $E^{\prime \prime }$ curves by means of simple normalization with respect to that found at C=0.5. The obtained single master curve is proposed in Figure 4.

In order to clarify the suggestions coming from the $E_{B}$ logistic function on the properties (dynamic end structural) of the studied SBR–CNT and SBR–CB mixtures, let us first consider, in Figure 5, the evolution in frequency of the two measured moduli, ($E^{\prime }$ and $E^{\prime \prime }$, at two extreme temperatures (i.e., 238 K (blue circles) and 303 K (red circles)); the plots are reported at increasing values of *C* in a clockwise way. The corresponding tanδ (asterisks) are also reported.

From the proposed data it is easily observable that the moduli (full circles are $E^{\prime }$ and open circles are $E^{\prime \prime }$) increase their absolute value by increasing *T* and *C*. A substantial difference in the frequency dependence between the behaviors of the moduli measured at low concentrations (C=0.5 and 1 phrCNT) and those at high concentrations (C>5 phrCNT) is evident. In the first case, all the reported quantities grow linearly with ω; in the second case (and at the highest 303 K), $E^{\prime }$, $E^{\prime \prime }$, and tanδ are practically constant. Such behavior is absolutely relevant because it proves the effects of the CNT on the system dynamics and that, in these conditions, our polymeric mixture is well inside the gel phase. Thus, at the highest concentration of nanotubes, the system is in a dynamic arrested phase while, at low *C* it is essentially a liquid.

This situation suggests the existence of a more structured phase in these thermodynamic conditions due to the effect of carbon nanotubes compared to what happens at low *T* and *C*. This is favored by the cross-linking between the chains of the SBR copolymer and CNT nanotubes, a process which, as we have said, gives rise to background contribution in the moduli.

In addition, the nanotube–rubber interactions and their local distributions affect the SBR molecular mobility with considerable influences on the collective properties of the mixture, whether mechanical or thermodynamic. Among the most sensitive will certainly be the transport and viscoelasticity functions. When mixed with other systems, CNTs have remarkable effects on the common resulting properties, including the $T_{g}$ value. Furthermore, SBR rubbers and CNT have corresponding glass transition temperatures that are quite different from each other: 250 K for rubber while that of CNTs is 338 K. Given this substantial difference between the respective $T_{g}$ (≃90 K), it is reasonable to expect that the $T_{g}$ of the mixture increases as the CNT concentration increases.

In this context, we believe that the data obtained by fitting the $E^{\prime \prime }$ by means of Equation (Equation 4) of the MCT can provide adequate information. The obtained values of the non-universal exponents *a* and *b* are reported in Figure 6 and, as can be observed, they show specific T−C dependence. Since these exponents reflect all the system properties being directly linked to the evolution of the loss moduli with ω and their maximum and minimum values at the different *T* studied (Equation (Equation 4)). They also provide an explanation to how CNT concentrations influence the system viscoelasticity and thus its dynamics. For both of these exponents, low CNT concentrations (0.5 and 1) have a different evolution in temperature from higher ones (C≥5). For the exponent *a*, although the numerical values are different, the *T* behavior is analogous: it is almost exponential for all *C* up to a certain temperature (T∼250 K), after which they tend to saturate towards a constant value. For the exponent *b*, sigmoidal behavior with *T* is observable: it is almost continuous for C=0.5 and 1, while for C≥5, marked discontinuity is evident at T∼250 K. On the one hand, such a result confirms the differences in the viscoelasticity of the system between the low and high concentrations of CNT. On the other hand, for the latter (C≥5), it represents the presence of some thermodynamic singularity strongly dependent on the temperature.

Such a hypothesis is fully confirmed from the thermal behavior of $E_{min}^{\prime \prime }$ and $\omega_{min}$, the other two quantities obtained from Equation (Equation 4) fitting. Their Arrhenius representation (log$E_{min}^{\prime \prime }$ vs. 1/T) is proposed in Figure 7 ($E_{min}^{\prime \prime }$ on the top and $\omega_{min}$ on the bottom). Also, in this case, different behaviors can be observed for the data thermal evolution at low and high CNT concentrations, respectively. For C≤1, both quantities are slightly different in their numerical value and exhibit the same super-Arrhenius behavior. At high concentrations, both propose a FSDC at approximately T=250 K, with the difference that, while the $E_{min}^{\prime \prime }\left(T\right)$ curves are *C* dependent, the corresponding $\omega_{min}\left(T\right)$ are *C* dependent only for temperatures below the crossover identified as $T_{L}$.

In the extended MCT form [23,24], the FSDC crossover temperature is identified as the critical $T_{c}$ of the ideal model [9], with $T_{L}$ located above that of the kinetic glass transition [13], i.e., $T_{L}>T_{g}$. Therefore, these latter results suggest that, at high CNT concentrations, the actual SBR-CNT system is explored in a temperature range in the vicinity of that of the dynamic arrest. This situation also shows the effects of carbon nanotubes on rubber $T_{g}$ by confirming what was observed in the physics of complex materials that collective phenomena like sol-gel or glass transition also depend on the system composition [5,6,7,9,23]. Specifically, in our case, this was observed by shifting this temperature to higher values proportional to *C*.

The MCT is based on precise scaling laws of the measured physical quantities with respect to the control parameter $\sigma =|T-T_{c}/T_{c}|$ under the universal assumption for the related exponents value. We used some of these to confirm the validity of such a hypothesis on the $T_{g}$ of the studied materials. Hence, we explored the thermal behavior of the obtained $\omega_{min}\left(T\right)$ and $E_{min}^{\prime \prime }\left(T\right)$ according to the corresponding MCT forms $\omega_{min}\approx {\left|\sigma \right|}^{1/2a}$ and $E_{min}^{\prime \prime }\approx {\left|\sigma \right|}^{1/2}$.

Using values at 248 K near $T_{L}$ and the transition temperature $T_{c}$, we checked the power laws for the two relevant parameters $\omega_{min}\left(T\right)$ and $E_{min}^{\prime \prime }\left(T\right)$. The corresponding data behaviors for C≥5 are proposed versus $|T-T_{c}/T_{c}|$ as a log–log plot in Figure 8. As can be observed, the two corresponding MTC scaling laws are well verified for that critical temperature. The related exponents are consistent with the MCT predictions on approaching $T_{c}$. In particular, we measured a≃0.35, whereas the predicted MCT value was 0.38.

This MCT scaling of $\omega_{min}\left(T\right)$ and $E_{min}^{\prime \prime }\left(T\right)$ represents new and important information on the dynamic properties of the studied SBR-CNT rubbers, showing us what happens to their kinetic glass transition. In addition, this is confirmation on the validity of the MCT approach not only for the analysis of viscoelasticity parameters for SBR-CNT rubbers in the region of the KGT but, in our opinion, also constitutes the proof that this theory represents a solid basis for dealing adequately with the complexity of the time–temperature superposition process.

## 4. Concluding Remarks

Using the MCT model developed for viscoelastic relaxation processes, we verified the TTS validity compared to the best known and most used approach based on the Williams–Landelp–Ferry equation (WLF) [16]. The main reason to use this statistical physics fundamental model developed to explore the dynamic arrest by means of the scaling concepts typical of the universal critical transition [30] is that we can clarify the chemical–physics processes underlying the TTS.

The studied system was styrene-butadiene rubbers (SBRs) filled with carbon nanotubes (CNTs) and carbon block (CB) whose elastic moduli were evaluated at different concentrations and temperatures in the region of the sol-gel transtion.

In this study, we considered (through the WLF) the measured frequency-dependent isotherms of the shear moduli $E^{\prime \prime }\left(\omega \right)$. The starting idea was that the susceptibility spectra χ(ω) of the relaxing system are characterized by an evolution with maxima and minima. For the MCT description of the viscoelasticity, the observed minima in loss moduli must scale according to the relation $E_{min}^{\prime \prime }\left(\omega \right)/E_{min}^{\prime \prime }$ versus $\omega /\omega_{min}$. In addition, MCT provides an interpolation form (equivalent to the main MCT form for correlators in the time regime) with which $E_{min}^{\prime \prime }$ and $\omega_{min}$ can be evaluated [26]. Through the fitting of our experimental isotherms, it was possible for us to evaluate both $E_{min}^{\prime \prime }$ and $\omega_{min}$ together with MCT exponents *a* and *b*.

Therefore, it was possible to operate the TTS by means of a multiplicative data shift of the obtained values of $\omega_{min}$ and $E_{min}^{\prime \prime }$. After which, we considered the $E_{B}$ background in the resulting master curves whose subtraction was evidenced for all curves frequency scaling ($\omega^{1/2}$) identical to the one suggested for the analysis of dielectric data [20]. In addition, these $E_{B}$ values have singular dependence on the filler (CNT) concentration, suggesting some dynamic effect due to the interaction between them and the SBR rubber.

Finally, considering this last fact, we studied the evolution in temperature of the obtained quantities from the fitting of the loss modulus isotherms. We first evaluated the T−C evolution of exponents *a* and *b*, observing that these quantities, especially *b*, show a significant dependence on the amount of CNT. In particular, *a* grew with continuity for all the studied *C* up to a certain temperature (T∼250 K), after which they tended to saturate towards a constant value. The exponent *b* instead showed (at about this temperatures) a marked discontinuity for C≥5, whereas for C=0.5 and 1, its growth was moderate and continuous.

Much more significant for the dynamics of the system were the evolutions with *T* of both $E_{min}^{\prime \prime }$ and $\omega_{min}$. By means of an Arrhenius plot of the respective data it can be clearly seen that the two lowest concentrations over the entire studied *T* range display super-Arrhenius behavior. The others are characterized by a crossover on purely Arrhenius behavior at about $T_{L}$. In agreement with recent experimental observations, theoretical models of statistical physics, and the MCT in its extended version, this is evidence of the fragile to strong dynamic crossover characteristic of kinetic glass transition that occurs near $T_{L}$ [13].

In conclusion, using MCT, this result is the most significant to treat the TTS of system response functions. In fact, it demonstrates that an interaction between the carbon nanotubes and the polymeric macromolecules of the rubber generates significant effects on the thermodynamics of the entire mixture. Specifically, it manages to change the temperature of its glass transition, bringing it to higher temperatures than that of the basic constituent (pure SBR).

As a result, the MCT scaling behavior of $\omega_{min}\left(T\right)$ and $E_{min}^{\prime \prime }\left(T\right)$ as a function of $|T-T_{c}/T_{c}|$ and the relative “critical” scaling as $\omega_{min}^{2\alpha }$ and $E^{\prime \prime 2}$ are well verified for the critical temperature $T_{L}=248$ K; in fact, the full value of the obtained exponents with the MCT predictions is α≃0.35, whereas the MCT predicted value was 0.38.

## Figures and Tables

**Figure 1 gels-10-00313-f001:**
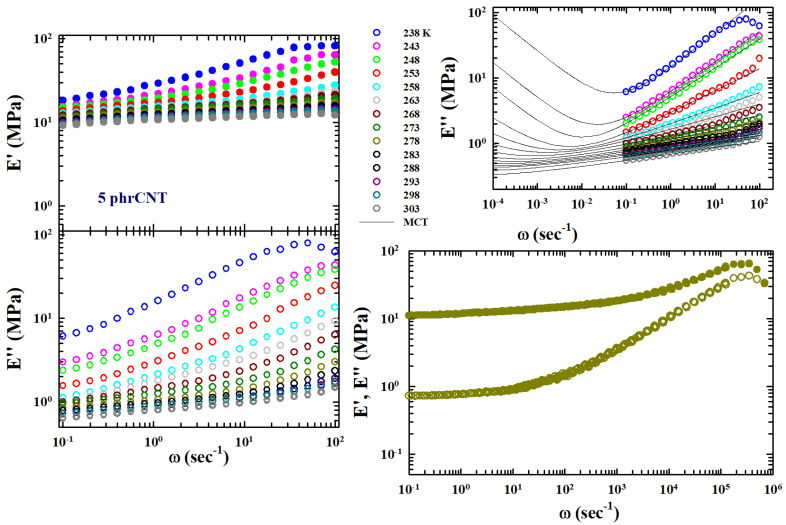
The tensile moduli, storage ($E^{\prime }$), and loss ($E^{\prime \prime }$) measured by means of the DMA in styrene-butadiene rubber filled with carbon nanotubes (CNT) at a concentration of C=5 (phrCNT, phr-parts per hundred rubber)) in a temperature range of 238–305 K, for frequencies (ω) from 0.8 to 100 ($\sec^{-1}$): left side. On the top of the right side, the data fitting of the loss moduli according to the MCT viscoelasticity model is presented. At the bottom, the corresponding MCT time–temperature superposition curves are presented for both $E^{\prime }$ and $E^{\prime \prime }$. All the data are represented in a log–log plot. Data were obtained from [25].

**Figure 2 gels-10-00313-f002:**
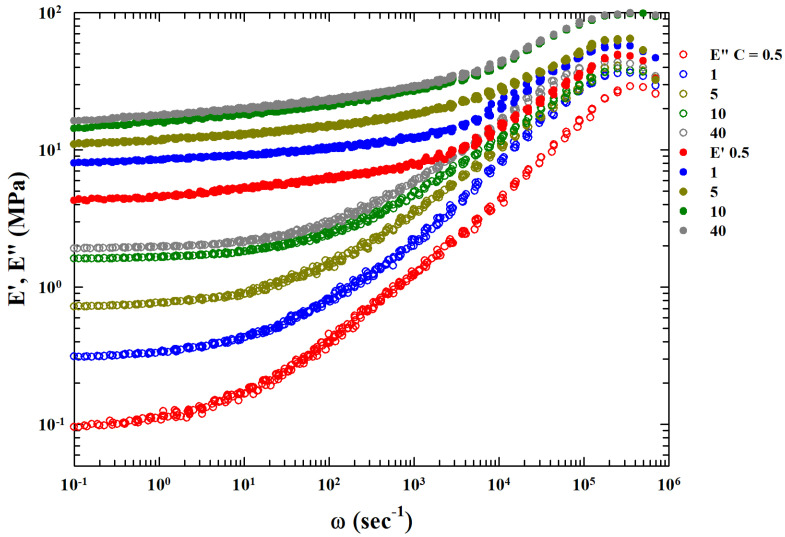
The MCT time–temperature superposition (TTS) curves for storage ($E^{\prime }$) and loss ($E^{\prime \prime }$) moduli for all the studied phrCNT concentrations C=0.5,1,5,10, and 40, presented in a log–log plot.

**Figure 3 gels-10-00313-f003:**
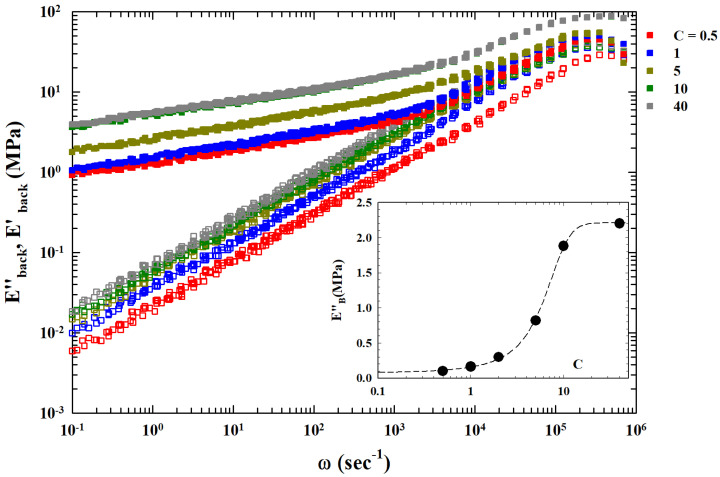
The TTS master curves after the subtraction of an extra background $E_{B}$. This assumes that $E_{B}$ is the asymptotic value of the moduli at low frequencies. The obtained $E_{back}^{\prime }$ and $E_{back}^{\prime \prime }$ are thus reported as fully and open symbols, respectively. Of particular interest in this log–log representation is the resulting loss moduli behavior: a power-law behavior (the straight lines) with a slope of approximately 0.5. In the figure inset is the proposed contribution $E_{B}^{\prime \prime }$ as a function of the concentration *C*, and the dotted line, as discussed in the text, is the data fitting in terms of a logistic function.

**Figure 4 gels-10-00313-f004:**
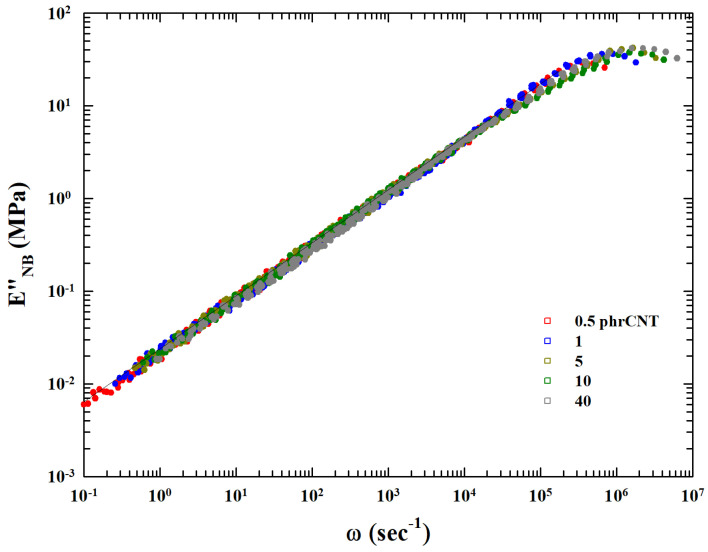
The TTF master curves representing the $E_{back}^{\prime \prime }$, superimposed as $E_{NB}^{\prime \prime }$, according to a simple procedure that considers the $E_{B}$ values.

**Figure 5 gels-10-00313-f005:**
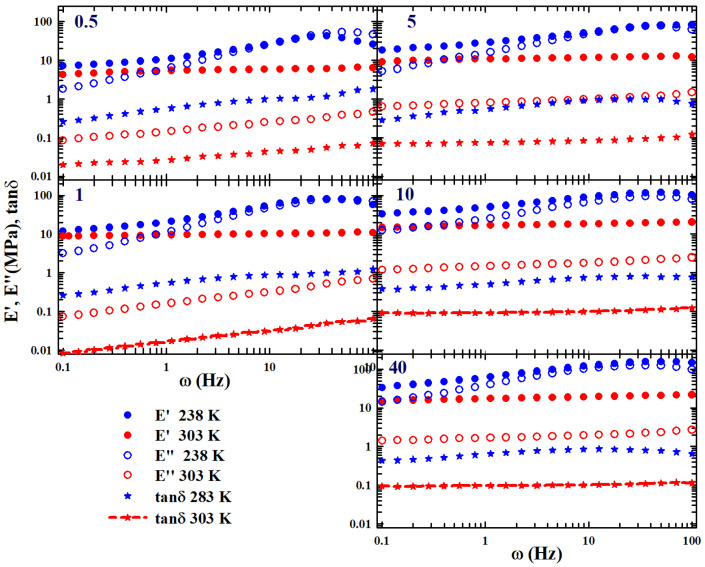
The figure reports (at increasing values of *C* in a clockwise way) the storage and loss moduli along with the loss angle δ tangent (tanδ), measured at two extreme temperatures, 238 and 303 K, for several different concentrations (0.5<C<40). From the illustrated data, it is easily observable that, at the highest temperature (303) and for C>5$E^{\prime }$, $E^{\prime \prime }$ and tanδ are essentially constant for all explored ω ranges.

**Figure 6 gels-10-00313-f006:**
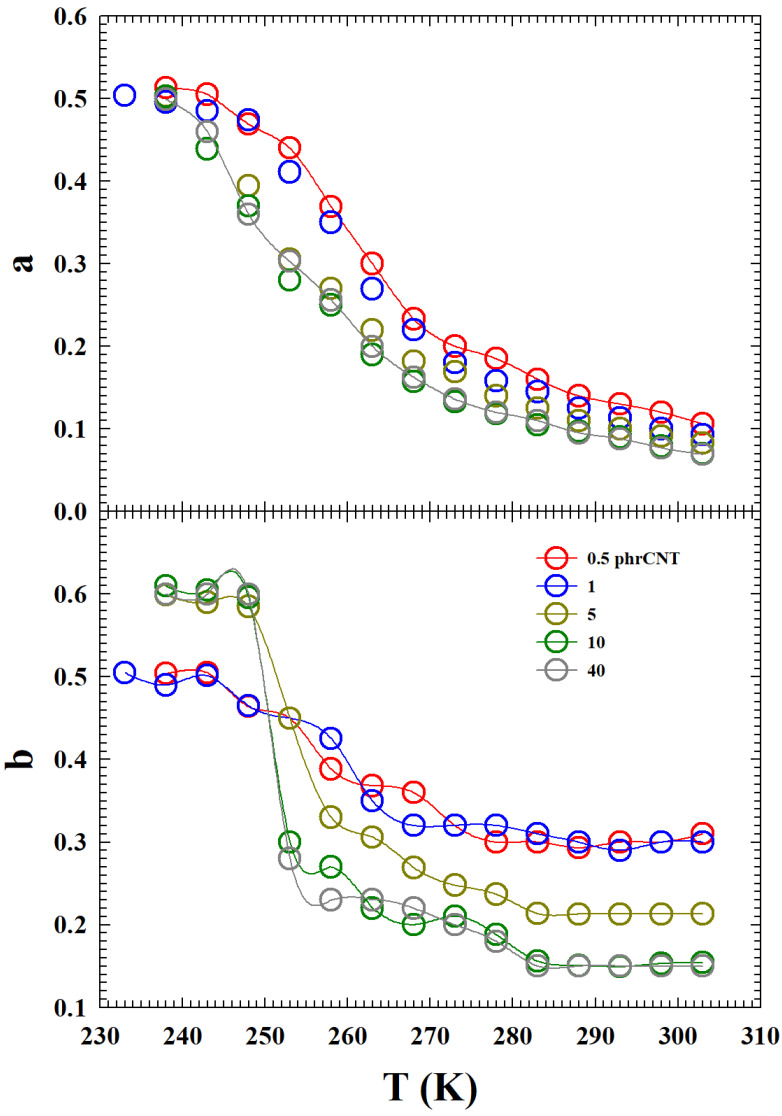
The T−C dependence of the MCT exponents (**a**,**b**) obtained from loss moduli data fitting (the lines are a guide for the eyes). *a* grows continuously (almost exponential) at all the concentrations up to a certain temperature (T∼250 K), after which they tend to saturate towards a constant value. *b* shows (around the same *T*), for C≥5, marked discontinuity.

**Figure 7 gels-10-00313-f007:**
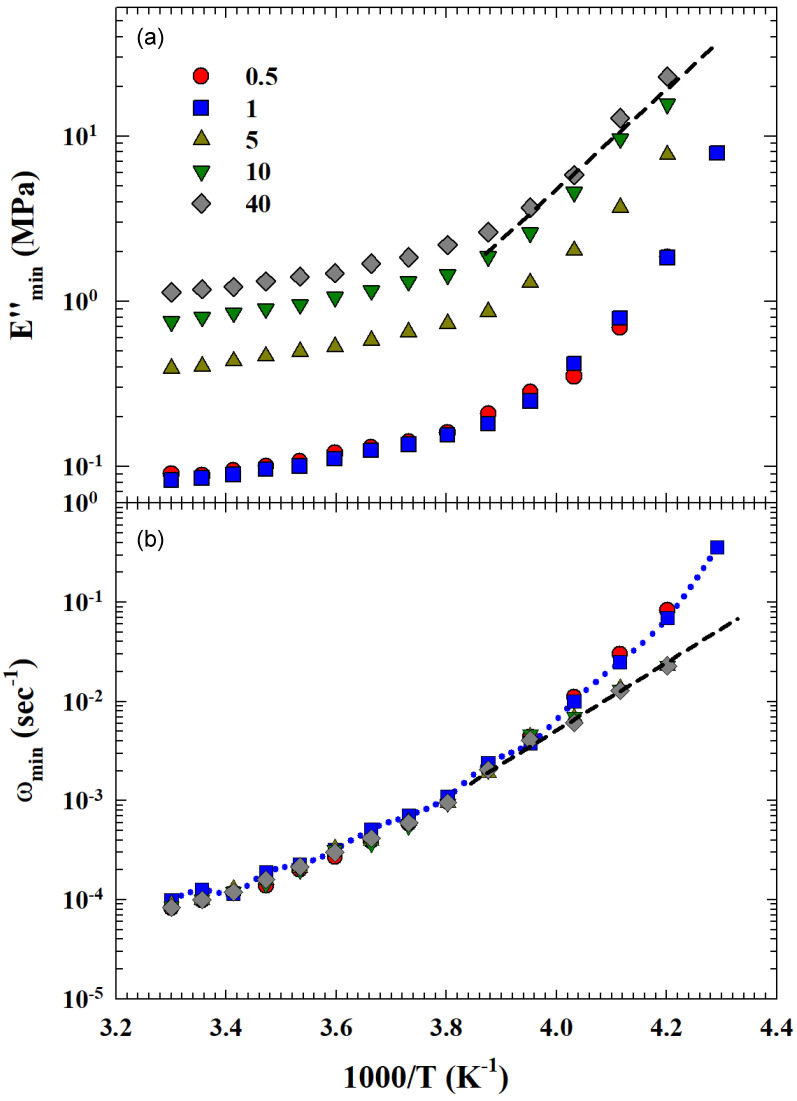
The Arrhenius representation of (**a**) $E_{min}^{\prime \prime }$ and (**b**) $\omega_{min}$, showing different T−C dependence. At the lower phrCNT (C=0.5 and 1), $E_{min}^{\prime \prime }$ and $\omega_{min}$ are nearly equal and exhibit super-Arrhenius behavior (dotted curve). For C≥5, a SFDC crossover can be observed at T≃250 K (dashed line). In this last case, $E_{min}^{\prime \prime }$, unlike $\omega_{min}$, show strong *C* dependence.

**Figure 8 gels-10-00313-f008:**
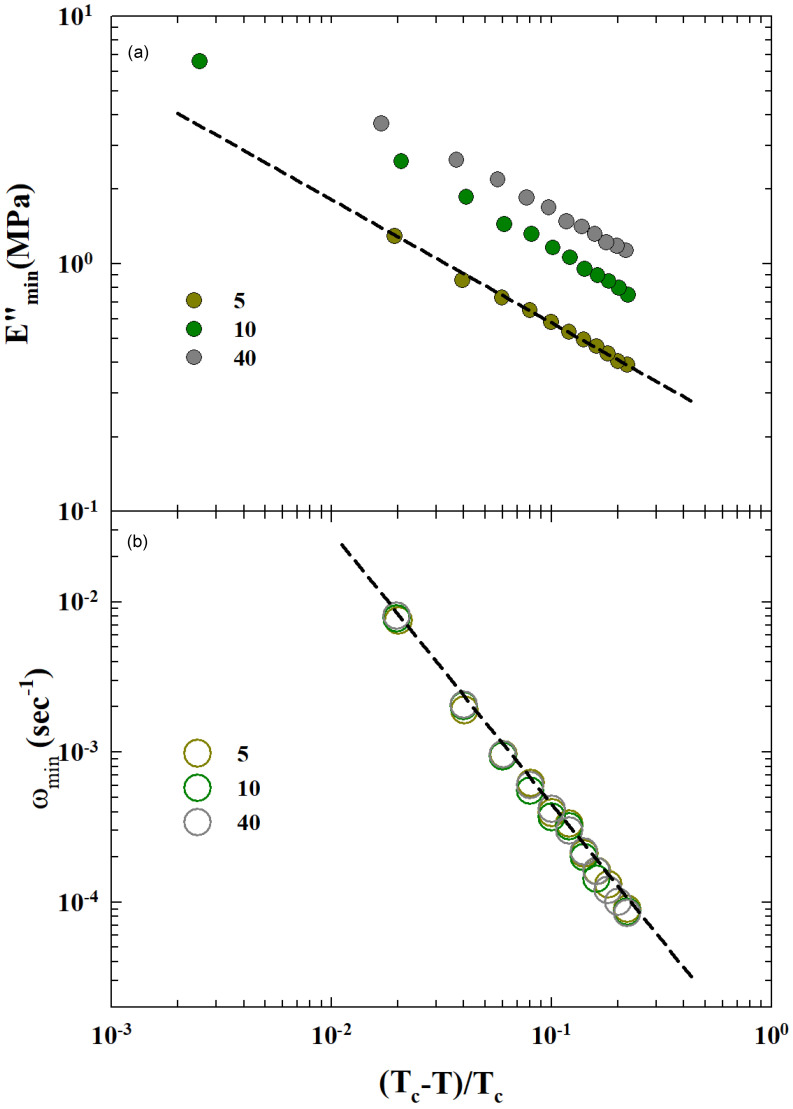
The critical temperature behavior of (**a**) $E_{min}^{\prime \prime }$ and (**b**) $\omega_{min}$. The value of the MTC critical temperature $T_{c}$ is for both quantities 248 K. The plotted dashed straight lines provide the corresponding critical index: ≃0.5 for $E_{min}^{\prime \prime }$ and ≃1.4 for $\omega_{min}$.

## Data Availability

The data that support the findings of this study are available from the corresponding author upon reasonable request.

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
