# Peer review of "The Time–Temperature Superposition of Polymeric Rubber Gels Treated by Means of the Mode-Coupling Theory"

_gels, 2024, doi:10.3390/gels10050313_

Round 1

Reviewer 1 Report

Comments and Suggestions for Authors

The authors investigated the validity of the time-temperature-superposition (TTS) approach to describe the viscoelastic behavior of styrene-butadiene rubbers doped with different amounts of carbon nanotubes. The hypothesis of a dynamic transition from brittle glass to solid glass, which depends on the amount of CNTs, was elegantly emphasized by applying the extended MCT theory.

The work as a whole is interesting and fascinating, both from a theoretical and practical point of view. The relevant conceptual and methodological points are well elaborated and discussed. However, some minor revisions could significantly improve the readability of the manuscript.  The specific points are:   

i)                    Revision of some sentences that are too long due to subordinate parts or information/comments added in brackets (e.g. lines 65-69, 149-153, 178-181, 187-192, 253-255, 273-276, 289-293

ii)                   Revision of punctuation in many places in the text: semicolon instead of colon, absence of the necessary colon, point instead of colon (lines 10, 21, 26, 44, 94, 127, 133, 138, 218, 261, from line 302 to line 304)

iii)                 The legend in Fig. 1 should indicate that the lower right shows the superposition curves for both E’ and E”

iv)                 Lines 201-122: The simple procedure by which EB values are taken into account to obtain the superimposed curves in Fig. 4 should be indicated.

v)                   Lines 230-231. It is more correct to say that for low concentrations on CNT both exponents….

Comments on the Quality of English Language

i)                    Revision of some sentences that are too long due to subordinate parts or information/comments added in brackets (e.g. lines 65-69, 149-153, 178-181, 187-192, 253-255, 273-276, 289-293

ii)                   Revision of punctuation in many places in the text: semicolon instead of colon, absence of the necessary colon, point instead of colon (lines 10, 21, 26, 44, 94, 127, 133, 138, 218, 261, from line 302 to line 304)

Author Response

We have considered with due care, point by point, all the sugges-
tions proposed. We thank the referee for the opinion given on our
work and for the related suggestions which allowed us to de ne a
more readable form suitable for the English language. Regarding the
qestion iv) we have alvo indicated the used procedure.

Reviewer 2 Report

Comments and Suggestions for Authors

Dear Authors,

I have completed the review of your manuscript titled "The Time Temperature Superposition in Polymeric Rubber Gels Treated by Means of the Mode-Coupling Theory". However, I would like to recommend some revisions before further consideration.

1)               You mention the use of the MCT model for viscoelastic relaxation processes instead of the commonly used Williams-Landel-Ferry equation (WLF). Could you please elaborate on why the MCT model was chosen over the WLF equation and how it provides a better understanding of the underlying chemical-physics processes of the time-temperature superposition (TTS)?

2)               You mention the presence of a background contribution (EB) in the resulting master curves. Please explain the origin of this background contribution and its relationship to the interaction between the CNT filler and the SBR rubber. How does the background contribution affect the TTS analysis?

3)               You discuss the dependence of the MCT exponents (a and b) on the CNT concentration. Please provide insights into the physical significance of these exponents and the observed dependence on the CNT concentration. How do these exponents relate to the dynamic properties of the system?

4)               You highlight a crossover from a super-Arrhenius behavior to a purely Arrhenius behavior for E"min and wmin at a certain temperature TL. Please explain the physical significance of this crossover and its relation to the kinetic glass transition. How does the presence of CNTs affect the glass transition temperature of the SBR rubber?

5)               Please provide more explanation on the scaling behavior and how it confirms the validity of the MCT approach. How do the obtained exponents compare with the predicted values by MCT?

Author Response

Since the referee's comments were very precise and timely and
therefore a stimulus to improve the work, we followed them carefully.
Regarding some di erent comments, 1) , 4) and 5) we have revised part of the introduction underlining that we used the extended MCT
model since, unlike the ideal one (and the TTS), it allows an accurate
determination of the properties of the system in the temperature
regions beyond the dynamic arrest.
We responded to comment 2) by proposing that the growth of the
CNT concentration leads to an increase in the polymer crosslinking
and therefore a development of local  uctuations. Phenomena that
are at the origin of the background contribution (EB).
Regarding the question 3) the meaning of the exponents a and b is
carefully described in paragraph 2.1 where their role in the response
functions that characterize the dynamics of the system is also ex-
plained (lines 121-133 of the manuscript). Furthermore, considering
the relevant and appropriate suggestion we have explained in the text
that their values are determined by the e ects of the CNT, at di er-
ent concentrations and temperatures, on the viscoelastic properties
of the materials studied.
As regards the  nal topic of question 4): our reply is reported at
the manuscript lines 273-277.

Reviewer 3 Report

Comments and Suggestions for Authors

reference 3. 8.12.13  the symble "&" between the page number appears messy codes.

ref 26 , The second name should be corrected.

ref 29 the style of the reference did not match the style of gels.

Comments on the Quality of English Language

There is no comment on the quality of English.

Author Response

Thanking the referee for his opinion, we have done what was re-
quested. In particular, correcting errors in the bibliography.

Round 2

Reviewer 2 Report

Comments and Suggestions for Authors

Dear Authors,

I would like to express my appreciation for taking my feedback into account and revising your work. Based on the improvements you have made, I am pleased to recommend the publication of your article in its current form.

Thank you for your hard work and dedication to this project.